# Therapeutic Strategies for Pancreatic-Cancer-Related Type 2 Diabetes Centered around Natural Products

**DOI:** 10.3390/ijms242115906

**Published:** 2023-11-02

**Authors:** Moon Nyeo Park

**Affiliations:** Department of Pathology, College of Korean Medicine, Kyung Hee University, Hoegidong Dongdaemungu, Seoul 05253, Republic of Korea; mnpark@khu.ac.kr

**Keywords:** pancreatic ductal adenocarcinoma, pancreatic stellate cells, pancreatic-cancer-related diabetes, natural product

## Abstract

Pancreatic ductal adenocarcinoma (PDAC), a highly malignant neoplasm, is classified as one of the most severe and devastating types of cancer. PDAC is a notable malignancy that exhibits a discouraging prognosis and a rising occurrence. The interplay between diabetes and pancreatic cancer exhibits a reciprocal causation. The identified metabolic disorder has been observed to possess noteworthy consequences on health outcomes, resulting in elevated rates of morbidity. The principal mechanisms involve the suppression of the immune system, the activation of pancreatic stellate cells (PSCs), and the onset of systemic metabolic disease caused by dysfunction of the islets. From this point forward, it is important to recognize that pancreatic-cancer-related diabetes (PCRD) has the ability to increase the likelihood of developing pancreatic cancer. This highlights the complex relationship that exists between these two physiological states. Therefore, we investigated into the complex domain of PSCs, elucidating their intricate signaling pathways and the profound influence of chemokines on their behavior and final outcome. In order to surmount the obstacle of drug resistance and eliminate PDAC, researchers have undertaken extensive efforts to explore and cultivate novel natural compounds of the next generation. Additional investigation is necessary in order to comprehensively comprehend the effect of PCRD-mediated apoptosis on the progression and onset of PDAC through the utilization of natural compounds. This study aims to examine the potential anticancer properties of natural compounds in individuals with diabetes who are undergoing chemotherapy, targeted therapy, or immunotherapy. It is anticipated that these compounds will exhibit increased potency and possess enhanced pharmacological benefits. According to our research findings, it is indicated that naturally derived chemical compounds hold potential in the development of PDAC therapies that are both safe and efficacious.

## 1. Introduction

Based on estimates, it is anticipated that PDAC, which represents the prevailing manifestation of pancreatic cancer, will ascend to the position of the second most prominent contributor to cancer-associated mortality on a global scale by the year 2030 [1,2]. Furthermore, the delayed detection of PDAC plays a crucial role in contributing to a worse prognosis. The prevalence of metastatic spreading and subsequent infiltration of important arterial structures has led to a significant observation that over 80% of cases lack the suitability for surgical removal of neoplastic growths [3]. Therefore, the discovery of new therapeutic approaches for PDAC is a pressing and unresolved issue in the field of medical research [4]. The gene known as Kirsten rat sarcoma virus (KRAS) is responsible for the synthesis of the Kirsten rat sarcoma viral oncogene homolog, a protein that plays a pivotal role in cellular signaling pathways. It is noteworthy to mention that mutations in the *KRAS* gene have been detected in roughly 90% of patients diagnosed with PDAC. The aforementioned alterations have an impact on both the control of gene expression and the amplification of gene copies. Furthermore, the onset of cancer requires the presence of genetic alterations in key genes like *TP53*, *CDKN2A*, and *SMAD4*, which are responsible for encoding the tumor suppressor protein p53, cyclin-dependent kinase inhibitor 2A, and SMAD family member 4, respectively [5,6]. In a subset of patients harboring the KRAS mutation, it has been observed that a phenomenon known as concurrent gene amplification occurs, affecting approximately 4% of these individuals [7,8,9]. The BRAF gene, which serves as the downstream signal to the KRAS gene, undergoes mutations in approximately 3–4% of cases independent of KRAS mutations [9,10]. The given text can be interpreted as a numerical representation. In the context of biology, it is important to note that whilst the majority of KRAS mutations lead to a state of constant activity, a minority of tumors harboring the KRASG12C mutation exhibit nucleotide cycling activity, which holds the potential for targeted intervention [11]. The identification of PDAC subtypes holds significant value in terms of prognostic implications and the potential for subtype-specific therapeutic approaches. However, it is important to note that laboratory models capable of accurately identifying each PDAC subtype require additional prospective validation before they can be routinely integrated into clinical practice [12]. The PDAC microenvironment is distinguished by an abundance of malignant epithelial cells, a substantial presence of stroma primarily composed of immunosuppressive T cells and myelosuppressive-type macrophages (M2), and a state of reduced vascularity [12]. Thus, PSCs have a pivotal function in the generation and renewal of the stroma. Upon stimulation by growth factors such as TGFβ1, platelet-derived growth factor (PDGF), and fibroblast growth factor, these cells exhibit the secretion of collagen and other constituents of the extracellular matrix [13,14]. The stroma, which is present in PDAC, a malignancy that originates in the pancreas, can comprise a substantial proportion, varying from 80% to 90% of the total tumor mass. The stroma has been observed to exhibit a partial association with the proliferation of PSCs. The pancreas is a complex organ in which PSCs play a crucial role as the main progenitors involved in the formation [15,16,17]. PSCs exhibit a state of inactivity in the normal pancreas and exhibit the presence of alpha-smooth muscle actin (α-SMA) upon activation, leading to their transition into a myofibroblast-like phenotype [18]. The activation of PSCs has a significant impact on the development of PDAC due to their increased production of growth factors and rich extracellular matrix [19,20]. Moreover, various stimuli encompass many states, namely, hyperglycemia, obesity, and hyperinsulinemia. Upon initiation, PSCs secrete specific signaling molecules referred to as cytokines, which are involved in facilitating diverse mechanisms within neoplastic cells. The aforementioned mechanisms encompass heightened cellular proliferation and invasion inside tumors, promotion of metastasis, initiation of epithelial–mesenchymal transition (EMT), and acquisition of resistance to chemotherapy [21,22,23,24,25,26,27,28,29,30]. The following section endeavors to provide a comprehensive overview of the current understanding pertaining to PDAC intricately associated with type 2 diabetes (T2D), leading to the hypothesis of PSC activation. This hypothesis was postulated to elucidate the mechanisms by which cancerous tissue contributes to the progression of disease.

### 1.1. PSC Activation Spurred Due to Type 2 Diabetes Has Become an Attractive Possibility for PDAC Stem Cells

Recent research has conducted an analysis on a specific group of PSCs, revealing a probable resemblance in biological characteristics to pancreatic stem cells [31]. PSCs have been identified to have persistent intercellular communication and to impact the formation of pancreatic cancer cells (PCCs). PSCs have emerged as a crucial element in the physiological processes of PCCs, playing a significant role in tumor advancement, evasion of immune response, and resistance to chemotherapy in PDAC [32,33,34,35]. Glutamine (Gln) plays a pivotal role as a vital nutrient in the process of carcinogenesis, serving as the primary provider of carbon and nitrogen for several metabolic activities [36]. It has been revealed that PCCs and PSCs have Gln metabolism crosstalk. PSCs exhibited a greater level of gene expression for glutamine synthetase (GS) compared to PCCs. Furthermore, the increased expression of GS, whether observed in the tumor or stromal cells, was found to be significantly correlated with an unfavorable prognosis in patients diagnosed with PDAC. Wnt initiates a direct connection with the β-catenin/TCF7 complex and binds directly to the GS promoter. The inactive form of GSK3β may activate the signaling pathway, increasing GS’s expression [37]. Notably, the crucial role of glutamine in modulating redox homeostasis through the synthesis of glutathione (GSH) and generation of nicotinamide adenine dinucleotide phosphate (NADPH) is evident in several populations of cancer stem cells (CSCs). Interestingly, studies have demonstrated that the disruption of glutamine metabolism effectively hinders the process of self-renewal and reduces the expression of genes associated with stemness and pluripotency, achieved by the elevation of intracellular reactive oxygen species (ROS) levels [38,39,40,41,42]. Moreover, IGF2 mRNA translation and mature secretion are fastest and most efficient at high glutamine and glucose concentrations. Controlled secretion secretes IGF2, but unlike insulin, the intracellular pool is insufficient to support secretion and requires constant synthesis. Additionally, glutamine-induced IGF2 release increases Akt signaling. Nutrient-regulated beta cell IGF2 synthesis triggers the beta cell bulk and function-controlling IGF2/IGF1R autocrine loop [43]. Adult tissue-specific stem cells, also known as somatic stem cells, belong to a distinct cell population that exhibits remarkable characteristics such as self-renewal and the capacity to undergo differentiation into specialized cell types with specific functions. Pancreatic stem cells have garnered significant attention in the realm of research over the past decade due to their developmental potential and the advantageous connections they share with other tissue-derived stem cells in terms of developmental biology and anatomy [44,45]. PSCs are a versatile and multifaceted cell population that can be observed in both the endocrine and exocrine components of pancreatic tissue. These cells make up approximately 7% of the total parenchymal cell population within the pancreas [46]. In recent studies, PSCs have emerged as a promising candidate for the generation of new β cells, which are responsible for insulin production in the pancreas [47]. PSCs have the ability to undergo activation, resulting in the acquisition of a myofibroblast-like phenotype characterized by the expression of α-smooth muscle actin (α-SMA) as an activation marker protein [48,49]. This activation process is accompanied by a decrease in the quantity of retinoid-containing fat droplets [15,50]. Importantly, PSCs are implicated in the pathogenesis of islet fibrosis, a process that significantly contributes to the development of β-cell dysfunction [51]. Activated PSCs exhibit a distinct expression of α-SMA and actively secrete collagen I, collagen III, fibronectin, and various other extracellular matrix (ECM) components. The information included within the aforementioned PSC is effectively depicted in Figure 1. This concerted effort by activated PSCs contributes to the facilitation of pancreatic fibrosis. The identification of lipid droplets, along with the concomitant expression of GFAP, nestin, desmin, and vimentin, serves as a defining characteristic for the quiescent phenotype of PSCs [52]. The current collection of evidence pertaining to the association between PSCs and islets has revealed that PSCs possess the ability to not only stimulate the development of fibrous tissue in islets but also play a role in the endocrine functions associated with pancreatic diseases like glucose intolerance, along with the preservation of islet viability [53,54,55]. PSCs are strategically positioned in close proximity to the basolateral aspect of pancreatic acinar cells. They are primarily found surrounding small pancreatic ducts and blood vessels within the pancreatic tissue [46]. According to research by Haraguchi et al. [56], side population (SP) cells from human cancer cell lines of different gastrointestinal tracts overexpress ATP-binding cassette super-family G member 2 (ABCG2), ATP-binding cassette super-family G member 1 (ABCB1), and CEA cell adhesion molecule 6 (CEACAM6), which are implicated in chemoresistance and have several traits of CSCs [57]. Hence, it is plausible that SP cells, nestin-positive cells, and PSCs could potentially occupy identical spatial positions. Taking into account the distinctive features encompassing the expression of nestin in SP cells and nestin-positive cells, it is plausible to posit that pancreatic stem/progenitor cells (PSCs) could potentially embody a comparable cellular entity within the adult pancreas, fulfilling analogous roles as stem/progenitor cells [58]. A recent discovery has been made regarding the presence of a specific group of cells expressing nestin in pancreatic islets and ducts [59]. The isolated nestin-positive islet-derived progenitor cells (NIPs), generated from adult pancreatic islets, have the ability to undergo differentiation in vitro, resulting in the development of cells exhibiting pancreatic exocrine, endocrine, and hepatic characteristics [60]. The existence of functional glucagon-like peptide-1 receptors (GLP-1Rs) on pancreatic progenitor cells implies a potential direct involvement of GLP-1R in the process of differentiating pancreatic progenitor cells into insulin-producing cells. Differentiation takes place within a specific fraction of NIP cells, which are stimulated to generate insulin. GLP-1R demonstrates a range of actions on β cells [61,62,63]. The expression and localization of the GLP-1R occurs in both normal pancreas and pancreatitis (AP/CP) tissues [64]. GLP-1R agonists have the potential to induce activation of PSCs located within the islets [65]. Homo-Delarche et al. have demonstrated the presence of PSCs within the islets of diabetes patients [66]. Chronic pancreatitis represents a prevalent risk factor for the development of pancreatic cancer, thus necessitating more scrutiny of the association between GLP-1R agonists and the induction of chronic pancreatitis [67,68]. Although GLP-1R agonists offer numerous benefits in the management of type 2 diabetes, it is imperative to no longer disregard the potential risk they pose in terms of pancreatic cancer development [69]. The available body of evidence indicates that there is a limited subset of cancer stem cells (CSCs) present among the PDAC cell population. CSCs are known to have significant involvement in the malignant characteristics of PDAC. Specifically, the identification of pancreatic CSCs is often achieved through the utilization of SP cells, the sphere-formation assay, and various CSC markers [70,71,72]. These elements collectively contribute to our understanding of the pivotal roles played by CSCs in the progression and behavior of PDAC [73]. Thus, PSCs exhibit the expression of many stem cell markers, possess multipotency, and demonstrate the capability to undergo effective differentiation into insulin-producing cells [74,75,76,77]. However, plasticity has the ability to safeguard CSCs and result in resistance. This is exemplified by the increased expression of enzymes and pathways involved in glutamine production, as well as the emergence of CSC subsets that are not reliant on glutamine. Plasticity facilitates the dynamic transition between CSC and non-CSC states, which are associated with distinct metabolic characteristics [78].

### 1.2. Type 2 Diabetes Is a Significant Prognostic Indicator for Pancreatic Cancer

According to epidemiological study, individuals diagnosed with type 2 diabetes (T2D) have an increased likelihood of developing several common types of cancer. This analysis focuses on T2D due to the lack of clarity regarding the potential cancer risk linked with type 1 diabetes (T1D). The efficacy of cancer treatments may be impacted by the presence of diabetes. The management of glucose is impacted by cancer and its treatment, and emerging evidence suggests that cancer itself can further complicate the care of individuals with diabetes [79]. There are several biochemical pathways that have been identified as connecting obesity to cancer. These pathways include insulin resistance and anomalies in the IGF-I system, production and transport of sex hormones, subclinical chronic low-grade inflammation and oxidative stress, and alterations in adipokine pathogenesis [80]. Adipose tissue hypertrophy, hyperplasia, or a combination of both can be induced by excessive caloric intake [81]. Hypertrophic adipocytes have the ability to generate and release proinflammatory adipokines and cytokines, which in turn attract macrophages [82]. The presence of elevated macrophages and inflammatory cells leads to the induction of both local and systemic inflammation through the upregulation of cytokine production, specifically interleukin-6 (IL-6) and tumor necrosis factor-alpha (TNF-α) [83]. The cytokines TNF-α have been found to interfere with the signaling pathway of insulin, leading to insulin resistance in adipocytes and an elevated production of free fatty acids (FFAs). The presence of elevated levels of FFAs in non-adipose tissues such as the liver, skeletal muscle, and pancreatic β cells has been found to be associated with the development of insulin resistance, steatosis, lipotoxicity, and metabolic dysfunction [84]. Notably, the phenomenon of intrapancreatic fat accumulation, which has a direct impact on the macro- and microenvironment of tumors, has been acknowledged as a potential indicator for predicting the extent of suffering experienced by patients with PDAC [85,86,87,88]. The production and release of insulin is exclusively attributed to pancreatic β cells, whereas the synthesis of IGF-1 primarily occurs in the liver. Both IGF-1 and insulin exhibit sequence similarities [89] and have the ability to trigger mitogenic pathways and inhibit apoptosis. The activation of the IGF-1–insulin pathway by obesity is known to induce intracellular signaling through MAPKs or the PI3K–AKT oncogenic cascade, as established in previous studies [89,90,91]. The activation of AKT and ERK signaling pathways by adipocytes leads to the occurrence of KRAS mutations in cancer cells. Leptin, adiponectin, lipocalin-2, and resistin are adipokines that exhibit high expression levels and play a significant role in fostering a proinflammatory milieu that promotes the advancement of cancer [92,93,94]. The available evidence indicates that individuals diagnosed with PDAC and T2D exhibit a higher presence of CD4+, CD68+, and CD8+T cells within their tissues. This implies that the adaptive immunological response of hyperinsulinemia may enhance the ability of tumors to evade the immune system and counteract immunosurveillance [95]. According to research findings, CD36 plays a function in that infiltrate tumors, resulting in lipid peroxidation. Consequently, the efficacy of interferon gamma (IFN-γ), a cytotoxic cytokine with anticancer properties, is diminished [96]. Compared to healthy controls, established T2DM patients had a distinct cytokine profile, but there was no discernible shift in lymphocyte subsets, which suggests immune system dysfunction [97]. Additionally, it has been observed that elevated levels of inflammatory cytokines are present in the early stages of T2DM and have the ability to predict the progression of this condition by reducing insulin sensitivity [98,99]. Islet amyloid polypeptide (IAPP) is synthesized along with insulin due to the presence of shared regulatory promoter sequences in the beta-cell-specific IAPP and insulin genes [100,101]. It is probable that the occurrence of hyperinsulinemia in situations of insulin resistance is accompanied by an elevation in quantities of serum islet amyloid polypeptide [102]. According to research, amyloid production increased with serum glucose–insulin ratio. Transgenic obese (ob/ob) mice that produced human islet amyloid polypeptide had higher serum glucose and lower serum insulin than non-transgenic mice, indicating that amyloid formation caused β-cell failure [103]. A recent discovery has demonstrated that glycosylation of human islet amyloid polypeptide enhances its propensity to form amyloid structures [104], suggesting that hyperglycemia may contribute to the development of islet amyloidosis by both stimulating the production of human islet amyloid polypeptide and augmenting its capacity to assemble into amyloid fibrils. The presence of this impact would not be observed in individuals with insulin resistance who do not have diabetes. Additionally, it is typically observed that these individuals do not possess islet amyloid, while having elevated levels of islet amyloid polypeptide in their bloodstream [105]. Additional research suggests that β-cell loss may be caused by different factors, such as endoplasmic reticulum stress, mitochondrial dysfunction, oxidative stress, inflammation, and islet amyloid buildup [105,106]. The research revealed a decline in cellular function, an increase in fasting blood sugar levels, and a moderate elevation in insulin resistance. Elevated levels of glucagon, somatostatin, and IAPP were seen in patients with PCRD. The restoration of hormone levels to a normal state was observed subsequent to the surgical removal of the tumor. This observation indicates that tumor cells have the ability to modify the amounts of glucagon, somatostatin, and IAPP [107]. The activation of PSCs occurs in response to inflammation and damage of the pancreas. Following the occurrence of an injury, immune cells are recruited to the affected site and subsequently secrete several cytokines, such as IL-1, TNF-α, and IL-6 [108]. Excess ECM synthesis in PDAC is caused by activated PSCs, and TGF-β1 is a crucial signaling element in this process [109]. TGF-β1 and FGF2 caused the increase in ECM protein synthesis, whereas PDGF most likely mediated PSC proliferation [109,110]. The endocrine pancreas is responsible for the production and secretion of many hormones, including pancreatic polypeptide, glucagon, somatostatin, and insulin, which are then released into the bloodstream [111]. The researchers discovered that around 52% of these patients had a recorded medical record of new-onset diabetes, which had been diagnosed within the 2–3 years leading up to the diagnosis of PDAC [112,113]. Patients with PCRD demonstrate a notable diminishment in the dimensions of islets, accompanied by a decline in the quantity of β cells [114,115]. Patients with PDAC may manifest the early signs of diabetes mellitus around 2 to 3 years before their official cancer diagnosis. The specific group of patients under consideration presents the opportunity for early identification and acknowledgment. The potential molecular associations between the mechanisms of carcinogenesis and diabetes may potentially offer a technique to identify early indicators of PDAC. Understanding the mechanics of PCRD holds promise in identifying biomarkers for the early identification of illnesses [114]. In order to gain a comprehensive understanding of the relationship between obesity and cancer, it is imperative to examine the alterations that occur in adipose tissue during weight gain. The signaling networks of insulin/insulin-like growth factor 1 (IGF-1) receptors have been implicated in the autocrine/paracrine promotion of many cancers, including ductal adenocarcinoma of the pancreas, which is recognized as one of the most fatal human diseases. There is an urgent need for the identification of novel targets for therapeutic intervention in pancreatic cancer [116]. Furthermore, this particular phenomenon possesses the inherent capability to unveil novel biological pathways, which may be strategically pinpointed for therapeutic interventions, ultimately resulting in improved patient outcomes.

### 1.3. Diabetes Management Strategies May Reduce the Risk of Pancreatic Cancer

Amyloid β (Aβ) is the constituent of the amyloid aggregates found in the brain affected by Alzheimer’s disease (AD) [117]. Conversely, the amyloidogenic peptide deposit in the pancreatic islets of Langerhans associated with diabetes is known as islet amyloid polypeptide (IAPP), which is a peptide consisting of 37 amino acids [118,119]. The observed IC50 values of flavonoids containing a catechol moiety were found to be lower compared to those lacking this moiety. This indicates that the specific arrangement and number of hydroxyl groups on the aromatic ring play a critical role in determining the strength of aggregation inhibition. The prevention of amyloid aggregation has the potential to be therapeutically beneficial in addressing age-related conditions such as Alzheimer’s disease and type 2 diabetes [120,121,122,123]. Phytochemicals, such as polyphenols and flavonoids, possess significant capabilities in the neutralization of free radicals and mitigation of oxidative harm, rendering them valuable in the management of DM [124]. The utilization of medicinal plant polyphenols has been found to have potential in reducing blood lipids and managing diabetes in individuals with diabetes. In a nine-month randomized, double-blind, placebo-controlled experiment (RCT) involving 240 individuals with prediabetes, it was observed that the group receiving curcumin did not develop type 2 diabetes mellitus (T2DM), while 16.4% of the participants in the placebo group did. Curcumin exhibited a protective effect on islet cells against the development of diabetes [125]. The anti-hyperglycemic effects of these substances are a result of their ability to competitively block α-amylase and α-glucosidase through binding to glucose transporters. By increasing the production of insulin from pancreatic B cells, attaching to receptors, reducing insulin resistance, and enhancing glucose tolerance, these plants may be able to combat diabetes. Improvements in glucose metabolism, B-cell size and activity, and plasma insulin, which decreases blood sugar, are further effects [126]. A randomized placebo-controlled trial was undertaken to investigate the impact of consuming an enriched bread containing a 1:1 combination of (-) epicatechin and quercetin on the anthropometric and biochemical parameters of the participants. After a period of three months, the intake of enriched bread on a daily basis led to significant reductions in total cholesterol, LDL (low-density lipoprotein)-cholesterol, total triglycerides, and fasting plasma glucose level [127]. The reduction of ROS formation by curcumin can be attributed to its impact on nicotinamide adenine dinucleotide phosphate (NADPH) oxidases, which leads to an increase in the activity of antioxidant enzymes. Additionally, curcumin is involved in the modulation of the Nrf2-Keap1 pathway [128,129]. The synergistic impact of swertiamarin and quercetin is believed to contribute to the heightened activation of viable β cells within the islets of Langerhans, leading to a more regulated secretion of insulin. The combination of swertiamarin and quercetin (CSQ) treatment substantially mitigated hyperlipidemia by reducing levels of total cholesterol, triglycerides, and LDL, while simultaneously increasing high-density lipoprotein (HDL) levels [130]. After conducting a comprehensive scientific study over a duration of ten years, researchers have observed a correlation between the prescription of metformin, an oral biguanide medication, and a reduced occurrence of cancer in individuals diagnosed with type 2 diabetes mellitus (T2DM). This observation was made when comparing these individuals to patients who were prescribed insulin or sulfonylureas [131,132]. Naringenin, a compound found in certain plants, has been observed to exhibit inhibitory effects on the process of gluconeogenesis. Additionally, it has been noted to stimulate the upregulation of adenosine-monophosphate-activated protein kinase (AMPK), a key enzyme involved in cellular energy regulation. These properties of naringenin have shown potential in the treatment of diabetes [133]. Numerous studies have provided evidence about the impact of antidiabetic medications (ADMs) on patient survival [134]. However, there remains ongoing discussion and uncertainty surrounding the potential effects of metformin or other ADMs on clinical outcomes specifically in patients with PCRD. The association between DM and unfavorable PDAC outcomes has been shown, leading to uncertainty regarding the impact of ADM on survival. Additionally, it was found that PDAC with DM had higher levels of CA19-9, a biomarker associated with this kind of cancer. Furthermore, the study indicated an inverse relationship between CA19-9 levels and prognosis in PDAC patients [134,135]. Notably, the manifestation of elevated CA19-9 levels was observed as early as two years before the clinical diagnosis. Significantly, CA19-9 has proven to be a valuable tool in providing crucial lead-time for the identification of resectable disease, wherein the implementation of multimodality therapy strategies can result in enhanced long-term survival rates [136,137,138]. As a result, individuals experience relapses due to the presence of cancer stem cells or the development of medication resistance, ultimately resulting in mortality. The early identification of pancreatic cancer is crucial in order to enhance the likelihood of survival. The present investigation into biomarkers in pancreatic cancer (PC) reveals that the serum carbohydrate antigen, CA 19-9, stands as the sole existing biomarker exhibiting an estimated specificity of 90% for PDAC [139]. Consequently, it is imperative to note that there exists a diverse array of pharmaceutical interventions that are considered essential for addressing the condition known as CA19-9. However, it is crucial to emphasize that additional inquiries and research endeavors are imperative in order to attain a comprehensive comprehension of the therapeutic effectiveness of these medications. The study indicated above suggests the potential presence of molecules with anti-type 2 diabetes mellitus (T2DM) properties in natural substances (Table 1).

### 1.4. Herbal Medicines In Vitro Assay to Improve PDAC Therapeutic Strategies

The current focus in the field of global medicine is directed towards the exploration and analysis of traditional medicinal practices prevalent in various regions, such as Korea, China, Japan, Thailand, and India. Traditional medicine holds a significant and prominent role within the realm of traditional medical practices [140]. Flavonoids that exhibit high activity levels frequently possess a catechol moiety, the functional group whose activity has recently been confirmed in other groups of polyphenolic compounds [141,142,143]. The extension of π-conjugation onto the carbonyl group in the C-ring, facilitated by the C2–C3 double bond, results in a greater radical scavenging capacity for unsaturated flavonoids compared to saturated structures like flavanones [143,144]. This particular study serves to emphasize the significance of the catechol moiety, a chemical structure commonly found in various biological compounds. Furthermore, a multitude of scientific investigations have provided evidence suggesting that the catechol moiety possesses the capacity to assume a crucial function in mitigating potential adverse effects associated with its utilization [145,146,147]. The utilization of antioxidant drugs serves as a viable approach to mitigate the detrimental effects of diseases that are instigated by oxidative stress. The catechol moiety, which is present in various antioxidants such as catecholamines and numerous flavonoids, plays a pivotal role as an antioxidant pharmacophore [143,148].

Large trees called *Abies spectabilis* can be found in Nepal’s Himalayan area at elevations between 3000 and 4000 m [149]. Thus, a phytochemical analysis of this bioactive extract was conducted, resulting in the isolation of ten compounds (**1**–**10**), one of which was a novel diterpene of the abietane type (1). Stems of *Abies spectabilis* (D. Don) Mirb. (Pinaceae) at 70% EtOH extract showed high cytotoxicity against MIA PaCa-2, with 100% cell death at 50 μg/mL [150]. Bioactive flavonoid apigenin has demonstrated significant anticancer potential. It has many biological features, including anti-inflammatory and anti-oxidant actions [151]. Apigenin increased the expression of apoptotic proteins and triggered the mitochondrial path of apoptosis for BxPC-3 and PANC-1, respectively. In BxPC-3 cells, apigenin significantly increased the expression of the cytokine genes IL17F, LTA, IL17C, IL17A, and IFNB1 [152]. A naturally occurring polyphenol obtained from plants, berberine (BBR), is found in many herbal medicines used in traditional medicine [153]. The inhibitory effect of BBR on cell proliferation was found to be more pronounced in BxPC-3 cells compared to HPDE-E6E7c7 cells. The administration of BBR resulted in a considerable increase in the activity of caspase-3 and -7 in both cell lines [154].

Dried turmeric rhizomes contain a hydrophobic polyphenol known curcumin, which is used in traditional herbal medicine and may one day be used as a cancer treatment drug [155]. Extract of the Coix seed, derived from the Coix lacryma-jobi (Yiyi ren) seed, has demonstrated efficacy as a cancer treatment in China, encompassing lung, pancreatic, and liver cancers. Co-treatment of two drugs was discovered to be more effective at sensitizing PANC-1 and BxPC3^luc^ cells to gemcitabine exposure. Tumor growth inhibition rate (TGI) in xenograft-null mice and BxPC3luc or PANC-1 cells treated with Coix seed extract showed a reversal of the elevated ABCG2 and ABCB1 protein levels caused by gemcitabine [156]. Curcumin had a significant inhibitory effect on PC cell metastasis through wound healing and the matrigel-transwell assay. The mechanism was TIMP1/TIMP2 overexpression accompanied by MMP2/MMP9/N-cadherin protein downregulation. When combined with either docetaxel or gemcitabine, curcumin demonstrated synergistic anti-cancer actions on PANC-1, HPAF-II, and MIAPaCa-2 cells [157]. *Eryngium billardieri* is a member of the Umbelliferae family, which is widely used as a medicinal plant throughout the world to treat a range of inflammatory diseases [158]. Bax could be induced by dichloromethane (DCM) and n-hexane (n-hex) extracts of *E. billardieri*, whereas cyclin D1 expression was reduced in PANC-1 and treated cancer cells, but had no effect on KDR/293 normal cells [159]. The alkaloid hernandezine (Her) is derived from *Thalictrum glandulosissimum*, a traditional herbal medicine. By increasing the phosphorylation of AMPK and decreasing the phosphorylation of mTOR/p70S6K, Her stimulated autophagy. Her presence increased reactive oxygen species (ROS) production in a concentration-dependent manner in PDAC cell lines Capan-1 and SW1990 [160]. One of the most common evergreen trees or shrubs, *Eucalyptus globulus* Labill., is endemic to Australia and Tasmania. The eucalyptus tree, a member of the Myrtaceae family, has become a global weed [161]. Alkaloids, polyphenols, and propanoids are just a few of the eucalyptus components that have been shown to have anti-cancer, anti-inflammatory, anti-fungal, anti-bacterial, and anti-septic effects [162,163,164]. The viability of PANC-1 cells was reduced, as demonstrated by the cytotoxicity of *E. globulus* extracts [165]. Black ginger, or *Kaempferia parviflora*, is a species whose rhizomes are a deep purple color. Its antiviral, antimycobacterial, antimalarial, and anti-ulcer properties have all been documented [166,167]. As a novel anti-austerity technique in cancer medication research, an agent that blocks pancreatic cancer cells’ resistance to food starvation and kills them preferentially in nutrient-starvation is being developed. With a IC_50_ of 3.3 µg/mL, *Kaempferia parviflora* exhibited strong anti-austerity efficacy against PANC-1 cells in addition to its ability to decrease colony formation [168].

As an essential component of tomatoes and other red veggies, lycopene is a potent antioxidant that has anticancer properties against a variety of cancers, including breast, pancreatic, prostate, and stomach cancers [169,170,171]. Pancreatic cancer PANC-1 cells undergo apoptosis as a result of lycopene’s suppression of NF-κB activation and reduction of ROS levels. Target genes such cIAP1, cIAP2, and survivin are downregulated as a result, although active caspase-3 and the Bax to Bcl-2 ratio are raised [172]. Qingyihuaji (QYHJ) is an herbal formulation developed by Professor Luming Liu. It consists of Banzhilian (Herba Scutellariae Barbatae, HSB), Baihuasheshecao (Herba Hedyotdis, HBHY), Tiannanxing (Rhizoma Arisaematis erubescentis, RAE), Jiaogulan (Herba seu Radix Gynostemmatis pentaphylli, HSRGP), and Doukou (Fructus Amomi Rotundus, FAR). This formulation has been extensively utilized in the treatment of prostate cancer (PC) over the past few decades [173,174]. A network was established to analyze the QYHJ target genes and PDAC target genes [175]. Through this analysis, 11 overlapped genes were identified, which is in partial agreement with our earlier research on immune-related targets of QYHJ in the context of PDAC [175]. In order to conduct further validation, a set of six key genes was chosen. These genes encompassed many biological processes, including inflammation (HMOX1, ICAM1, VCAM1, and CCL2), oxidative stress (NQO1), and apoptosis (Bcl2) [176]. A schematic demonstrating the network of protein–protein interactions (PPI) was created using the 26 possible target genes associated with the anti-prostate cancer (PC) effects of Xiang-lian pill (XLP). The findings indicate that MMP9, CASP8, CASP3, HSP90AA1, CTSB, MMP2, PTGS2, CASP9, IL4, and CTSD are the primary proteins involved in the anti-prostate cancer effects of XLP. These proteins are likely to have crucial functions and can be considered as potential target genes for the active ingredients of XLP [177]. The *Oxalidaceae* family is distinguished by its sour flavor and has a well-established history as a medicinal plant. It is recognized for its composition, which includes oxalic acid, malic acid, and tartaric acid [178]. The study revealed that *obtriangulata* methanol extract (OOE) exhibited anti-cancer properties when tested on BxPC3, a cell line associated with pancreatic cancer. The OOE treatment modified the activation of ERK, Src, and STAT3 signaling pathways, thereby regulating the expression of STAT3-downstream genes that are associated with tumor formation. Additionally, the impact of OOE on cell viability, proliferation, and the induction of apoptotic effects and accumulation at the G2/M phase in BxPC3 cells was determined [179]. In Asia, the lamiaceae family plant *Orthosiphon stamineus (O.s)* has a long history of traditional use in the treatment of numerous chronic diseases, including cancer [180]. Rosmarinic acid, eupatorin, sinesitin, pentacyclic triterpenes, betulinic acid, oleanolic acid, ursolic acid, and β-sitosterol are just some of the more than 20 phenolic bioactive components found in O.s leaves, according to phytochemical analyses [180,181]. *Et. Os* inhibited PANC-1 migration and activated caspase-3 in PANC-1 to induce early apoptosis via the cleavage of PARP proteins when combined with gemcitabine [182]. A traditional herbal medicine for ischemic stroke and cardiovascular disease uses Panax notoginseng saponins (PNS). PNS and gemcitabine (Gem) were given to MIA PaCa-2 and PANC-1 cells. Cell viability was measured by CCK-8, cell proliferation by colony formation and EdU, cell migration and invasiveness by wound healing and transwell, and cell apoptosis by flow cytometry [183]. *Portulaca oleracea* L. (*P. oleracea*) is a potent natural medicine that grows spontaneously and is planted worldwide. Pectin, carbohydrates, protein, fatty acids, notably omega unsaturated acids, and alkaloids are found in this plant. It contains vitamins A, C, and E, as well as minerals Fe, Cu, K, and Se [184,185,186]. The PANC1 cell line was subjected to apoptotic induction using *P. oleracea* plant extract, and the alterations in cyclin-dependent kinases (CDKs) and P53 gene expression were also examined. *P. oleracea* was more prevalent in those of the control HUVEC cells. Additionally, compared to the HUVEC cells as a normal cell line, there was a modest decrease in CDK1 expression in the cells treated with plant extract [187]. The anti-cancer properties of pomegranate (*Punica granatum*) have been attributed to its substantial content of flavonoids and polyphenols, such as ellagic acid, ellagitannins, quercetin, kaempferol, and luteolin glycosides [188]. Using a chick chorioallantoic membrane (CAM) model, the antiangiogenic effect of pomegranate was observed on human pancreatic cancer (Suit-2) cell lines. Additionally, the extract was found to have an effect on fibroblast growth factor (FGF2). Pomegranate extract dramatically decreased the hemoglobin concentration and tumor weight in pancreatic Suit-2 and colon Col205 CAM models [188]. The two flavonols that are most commonly found are quercetin and kaempferol [189]. Quercetin has been shown to have cytotoxic, antioxidant, hepatoprotective, antifungal, and anticancer properties [190]. The expression of N-cadherin, MMP-9, and STAT-3 signaling pathways is suppressed by quercetin, which may have the potential to limit epithelial–mesenchymal transition (EMT), invasion, and metastasis in PANC-1 cells [191]. *Rhus verniciflua* Stokes has been extensively utilized in traditional Eastern Asian medical systems for an extended amount of time [192,193]. It was discovered that downregulation of DR3 in PaC cells decreased the expression of pIkBalpha/beta kinases (pIKKs), MMP9, and XIAP, which are primarily responsible for conferring chemoresistance in PaC cells [194].

*Scutellariae baicalensis* Georgi’s dried root, or *Radix Scutellariae* (RS), is a popular herbal remedy in many Asian nations, used to cure cancer, hypertension, inflammation, and cardiovascular disease [195]. The flavonoid glycoside was hydrolyzed to aglycones to increase its bioavailability. Next, ethyl acetate was used to extract the total flavonoid aglycone extracted (TFAE) [196,197]. Autophagy is induced by TFAE via inhibition of the PI3K/Akt/mTOR signaling pathway. Furthermore, the caspase signaling pathway was utilized to enhance TFAE-mediated apoptosis in the number of annexin V/PI-positive cells and the cleavage of caspase proteins in BxPC3, and it demonstrated minimal cytotoxicity towards normal cells HPDE6-C7 cells [195]. Zicao (*Lithospermum erythrorhizon* or *Arnebia euchroma* (Royle) Johnst) is a traditional herbal medicine that has been shown to have blood-circulation-boosting and detoxifying properties. Its main active constituent has been extracted from the herb [198]. Shikonin exhibited a more pronounced kinase inhibitory effect against PAK1 inhibitors, like CP734, as well as cytotoxicity towards BxPC−3 and PANC−1 cell [199]. A perennial twining shrub of the Celastrales, *Tripterygium wilfordii*, has been utilized medicinally for centuries, primarily to treat rheumatoid arthritis [200]. Two of these natural compounds, celastrol and triptolide, are among five that have been recognized for their significant potential in the development of pharmaceuticals [201]. Triptolide eliminates cancer cells in the pancreas via two distinct mechanisms. It elicits autophagic death in metastatic cell lines S2-013, S2-VP10, and Hs766T while inducing caspase-independent apoptotic death in MiaPaCa-2, Capan-1, and BxPC-3 [202]. Thymoquinone (Tq), the most prevalent component of Nigella sativa seed’s essential oil extract, possesses antioxidant and anti-inflammatory properties through unknown mechanisms [203,204,205]. In MiaPaCa-2 and AsPC-1 pancreatic ductal adenocarcinoma (PDAC) cell lines, thymoquinone decreased proliferation and cell survival, produced partial G2 cycle arrest and SubG0/G1 arrest, increased p21 mRNA expression, upregulated p53, and downregulated HDAC activity. Moreover, Bcl-2; HDAC 1, 2, and 3; and H4 acetylation were also triggered. Thymoquinone downregulated Bcl-xL, Bcl-2, and XIAP and upregulated and activated pro-apoptotic molecules like caspases-3 and -9, Bax, cytochrome c release, inhibitory tumor growth, Notch1, NICD, PTEN, and Akt/mTOR/S6 signaling [206,207,208,209]. Wogonin (WOG), wogonoside, baicalein, and baicalin have been identified as bioactive compounds present in the roots of this particular plant [210]. Naringenin, a polyphenolic flavanone compound, is abundantly present in various medicinal plants as well as citrus fruits such as grapefruits, tomatoes, and cherries. Apoptosis was induced in SNU-213 cells by the upregulation of ASK1, P38, P53, JNK, and ROS [211]. The study demonstrated that the administration of naringenin resulted in a decrease in gemcitabine resistance and a reduction in cancer cell invasion in the AsPC-1 and PANC-1 pancreatic cancer cell lines [212]. The investigation of combination therapy has been conducted with the aim of mitigating medication resistance and enhancing treatment effectiveness. The concurrent treatment of naringenin and hesperetin in BALB/c nude mice resulted in a greater inhibition of cell growth, invasion, and p38 activation in Miapaca-2, PANC-1, and SNU-213 cell lines compared to the individual administration of either drug [213]. Wogonin (WOG), wogonoside, baicalein, and baicalin are bioactive chemicals in this plant’s roots [214]. PANC-1, Colo-357, and HPCCs4 human pancreatic cancer cells produced ROS after WOG treatment. Beclin-1/PI3K and Akt/ULK1/4E-BP1/CYLD signaling pathways were activated while mTOR was inhibited. WOG and other natural flavones increased p53 expression in Capan-1 and Colo-357 pancreatic cancer cells. These cells also inhibited Mcl-1, CDK-9, c-FLIP, and MDM2 expression [215,216]. *Ziziphus nummularia*, a prickly shrub in the Rhamnaceae family, is also referred to as Sidr [217]. *Ziziphus nummularia* is rich in bioactive compounds, including tannins, flavonoids, steroids, glycosides, and alkaloids. As a result, it has been used to cure a variety of pathological disorders, such as inflammation, bronchitis, diarrhea, anemia, and the common cold [218]. *Ziziphus nummularia* ethanolic extract (ZNE) inhibits Capan-2 pancreatic cancer cell proliferation, caspase-3-dependent apoptosis, migration, and invasive potential by downregulating MMP-9. ZNE inhibits ERK1/2(MAPK) and NF-κB pathways, reducing Capan-2 adhesion to collagen, integrin α2 expression, VEGF production, and angiogenesis [219].

The utilization of natural chemicals is restricted by their inherent low bioavailability in the absence of structural modifications, hence posing challenges in their application [214]. However, it is important to note that natural product-based herbal medicine is currently being recognized as a groundbreaking therapeutic strategy for a multitude of diseases, including cancer. This is primarily due to its exceptional effectiveness and the limited occurrence of adverse reactions [220]. Hence, one can posit that natural products are poised to occupy a central role as a groundbreaking therapeutic approach for the treatment of cancer in the upcoming decade [221]. The aforementioned discovery implies the plausible existence of anti-cancer attributes within natural compounds, thereby promoting a sense of motivation to delve deeper into this field of scientific investigation (Table 2).

### 1.5. Potential New Treatments Include Natural Compounds for Fighting PDAC

Several studies have been conducted, including clinical trials, to assess the effectiveness and safety of natural therapy for PDAC. The phenomenon of medication resistance, whether it is primary or acquired, has the potential to manifest both on-target and off-target [223]. The primary molecular target mutation of a medicine leads to on-target resistance, resulting in a reduction or complete loss of therapeutic effectiveness. Rapid and complex alterations by epigenetic modifications of tumor gene expression profiles contribute to the development of resistance to treatment [224]. In clinical practice, it has been found that the standardized allergen-removed RVS (aRVS) extract stabilizes the progression of cancer [225,226,227]. Patients with advanced pancreatic cancer who are unable to receive conventional therapy tolerate aRVS treatment well. Either alone or in conjunction with chemotherapy, aRVS may extend overall survival [228]. Coix seed extract, derived from the Coix lacryma-jobi (Yiyi ren) seed, has demonstrated efficacy as a cancer treatment in China [229].

European mistletoe (*Viscum album*) is commonly employed in combination with traditional cancer treatments or as a stand-alone alternative therapy for individuals diagnosed with cancer [230]. Mistletoe extracts include lectins, viscotoxins, flavonoids, and membrane lipids. Lectins tend to induce apoptosis and enhance the immune system [231,232]. Recent research has revealed potential implications of mistletoe in the prognosis of individuals diagnosed with colorectal or pancreatic carcinomas [233,234]. The findings of a randomized controlled trial demonstrated a significant increase in overall survival, from 2.7 months to 4.8 months, among patients diagnosed with advanced pancreatic cancer who were not eligible for antineoplastic therapy. This improvement in survival was shown in individuals who received subcutaneous mistletoe treatment, as compared to those who received the standard of supportive care [233,235]. QYHJ has been previously utilized as a supplemental medication in the treatment of PDAC, resulting in enhanced one-, three-, and five-year survival rates among patients, without any apparent adverse effects [236]. Nevertheless, the mechanism by which QYHJ affects PDAC is yet to be fully understood, perhaps due to the intricate nature of the substances involved and their interacting effects within traditional Chinese medicine (TCM) formulae. Therefore, it is imperative to conduct comprehensive pharmacological investigations in the future, which will involve the examination of various combinations and concentrations [176]. The administration of Xiang-lian pill (XLP) resulted in the inhibition of PC proliferation, while exhibiting little hepatotoxic and renal toxic effects. The pharmacological effects of XLP on PC encompass the modulation of many targets that operate within fat-metabolism-related signaling pathways. These effects have significant potential in the regulation of aberrant tumor metabolism. There is a need for conducting extensive randomized controlled studies in order to provide a more robust body of data for the potential utilization of XLP in this setting of PDAC [177]. The administration of a combination of curcuminoids and gemcitabine for a median duration of two weeks in individuals diagnosed with advanced pancreatic cancer did not result in a statistically meaningful therapeutic outcome [237]. Hence, the use of a composite of pharmacological substances, each specifically formulated to target separate pathways, has the potential to provide a synergistic or potentiation effect. It is interesting to acknowledge that both of these occurrences have the potential to yield significant benefits in the context of cancer treatment. Individuals diagnosed with various types of solid tumors, such as breast, colorectal, and stomach cancer, were recruited to participate in a phase II clinical investigation that utilized a double-blind randomized trial methodology. The researchers made a significant observation in which the addition of curcuminoids to chemotherapy led to a decrease in negative side effects and an improvement in the general health of the participants in the study [238,239,240]. Minnelide, a water-soluble derivative of triptolide, has demonstrated efficacy in the treatment of several malignancies through multiple clinical trials [241]. Minnelide exhibits a synergistic effect when combined with gemcitabine and nab-paclitaxel, both of which are chemotherapeutic drugs currently licensed for pancreatic cancer treatment. This synergistic effect has been observed in several models of pancreatic cancer [242]. A Chinese open-label, randomized, multicenter trial comparing gemcitabine monotherapy versus Kanglaite injection (KLTi) plus gemcitabine as first-line treatment for advanced pancreatic cancer showed favorable outcomes [243]. KLTi in combination with gemcitabine was compared to gemcitabine monotherapy in US patients with advanced pancreatic cancer in this phase 2b research [229]. PDACs patient exhibit a remarkably low sensitivity, with a response rate of less than 5%, to immune checkpoint inhibitors [244,245]. The limited infiltration of T cells in PDAC can be partially attributable to the low number of mutations present in the tumor cells. This results in a dearth of neoantigens, which are molecules that can be recognized by the immune system and trigger strong T-cell responses [245,246]. Neoantigens are proteins that are absent in healthy tissues and serve as markers for cancer cells, signaling their foreign nature to T cells. Consequently, PDACs may exhibit weak antigenicity and have a reduced number of infiltrating T cells [247]. CD3+ T cells are of paramount importance in the cellular immune response of the host and can be categorized into two subtypes: CD4+ helper T cells and CD8+ cytotoxic T cells. The immune responses are enhanced by these cells through the production of lymphatic factors by CD4+ helper T cells, thereby stimulating additional lymphatic cells that are essential for exerting an antitumor effect [248]. Interestingly, there have been reports indicating that Korean red ginseng possesses immune-modulating capabilities, which may lead to an enhancement in T-cell proliferation. This suggests that it has the potential to modulate cellular immunological responses [249,250]. The study revealed that the concurrent administration of Korean red ginseng alongside adjuvant chemotherapy led to an increased count of CD4+ lymphocytes and a higher CD4+/CD8+ T lymphocyte ratio following chemotherapy in individuals diagnosed with bile duct or pancreatic cancer. There was no significant difference observed in the prevalence of neutropenia and liver dysfunction between the groups [251]. In addition to enhancing therapeutic efficacy, the utilization of a drug–herb combination concurrently diminishes the dosage magnitude of each individual pharmaceutical agent that is administered. One plausible therapeutic strategy involves the immunomodulation of malignant tumors in PDAC through the activation of CD4+/CD8+ T cells. These specialized T cells possess the ability to identify and target neoantigens, which are unique antigens found on cancer cells. The aforementioned environment may be effectively addressed by the utilization of a combined approach involving conventional drugs, hence leading to the attainment of a potent antitumor immunotherapeutic response. The aforementioned clinical trials associated with natural substances are delineated in Table 3, thereby indicating the potential existence of anti-carcinogenic attributes. This, in turn, provides as a motivation for the pursuit of additional investigations within this field of knowledge.

### 1.6. Tendency in FDA-Approved Therapies

FOLFIRINOX (leucovorin, fluorouracil, irinotecan, and oxaliplatin) is advised for patients with an Eastern Cooperative Oncology Group (ECOG) performance status (PS) of 0 to 1, favorable comorbidity profile, patient preference, and a support system for aggressive medical therapy [252]. Patients who meet the criteria of receiving first-line gemcitabine plus NAB-paclitaxel, including a relatively favorable comorbidity profile, patient preference, and a support system for aggressive medical therapy, as well as having access to chemotherapy port and infusion pump management services, may be eligible for second-line therapy with 5-FU plus oxaliplatin, irinotecan, or nanoliposomal irinotecan. Gemcitabine or fluorouracil may serve as viable options for persons with cancer and comorbidities that restrict the use of more intensive treatment regimens when employed as second-line therapy [252,253,254,255]. The utilization of nanoliposomal irinotecan (nal-IRI) alongside 5-FU and LV has been recognized as a viable approach in the treatment of PDAC cases that have developed resistance to gemcitabine-based therapy [256]. The therapeutic interventions available for individuals diagnosed with advanced pancreatic cancer (APC) encompass a spectrum of approaches, including the use of gemcitabine as a single agent or the implementation of multi-drug treatment protocols. The selection of a specific therapeutic option is contingent upon various factors such as the patient’s age, performance status, presence of other medical diseases, and the preferences of both the patient and their healthcare provider [257]. The treatment landscape for metastatic pancreatic cancer becomes increasingly uncertain beyond first therapy since patients frequently experience rapid clinical decline and are no longer eligible for further interventions beyond optimal supportive care [258]. Through a comprehensive analysis of publicly accessible bioinformatics data, it has been determined that the administration of FOLFIRINOX has a regulatory effect on tumor immunity. Herein, the crucial protein C-X-C motif chemokine 5 (CXCL5) was discovered, which was found to produce an immunosuppressive milieu in PDAC [259]. The gene CXCL5 is classified as a member of the C-X-C chemokine family, which is widely recognized for its role in attracting granulocytic immune cells by specifically binding to its receptor, C-X-C chemokine receptor type 2 (CXCR2) [260]. The significance of CXCR2 lies in its involvement in the recruitment of tumor-associated neutrophils (TANs), together with CXCL8, CXCL6, and CXCR1. These factors collectively contribute to the suppression of anticancer immune responses [261]. Novel biomarkers, such as KRAS mutations, NTRK1–3 fusions, and BRCA1/2 mutations, have demonstrated potential for targeted therapy in individuals with PDAC. The medications entrectinib, larotrectinib, and olaparib continue to be limited in their application and demonstrate suboptimal efficacy [262]. The inhibition of CXCL5 may present a promising option for targeted therapy, and conducting additional research on CXCL5 could facilitate the integration of FOLFIRINOX and immunotherapy [259]. In the field of PDAC, it has become customary to employ maintenance therapy subsequent to the administration of initial combination cytotoxic therapy for specific cohorts of patients. Arm A involved a six-month administration of FOLFIRINOX, while arm B received four months of 5-FU/LV maintenance therapy. Lastly, arm C underwent treatment with gemcitabine and FOLFIRI (5-FU, folic acid, irinotecan) on a bi-monthly basis. The investigation sought to assess the prospective efficacy of maintenance chemotherapy. In arms A, B, and C, the six-month progression-free survival (PFS) rates were observed to be 47%, 44%, and 34%, respectively. Additionally, the median overall survival (OS) durations were found to be 10.1, 11.2, and 7.3 months in arms A, B, and C, respectively [12,263]. The coadministration of gemcitabine with the FOLFIRINOX regimen, consisting of folinic acid, fluorouracil, irinotecan, and oxaliplatin, has been found to augment its efficacy in treating metastatic pancreatic cancer. However, this therapeutic approach has been associated with a notable rise in the incidence of side events and toxicities [257]. As a result, the combination of selective inhibitors and chemotherapeutic agents is increasingly being utilized as a method to alleviate the detrimental impacts on the human body [264,265]. The approval of Olaparib by the FDA is for the purpose of maintenance therapy in patients with PDAC who possess germline BRCA1/2 mutations [266]. The presence of wild-type KRAS in PDAC individuals signifies a distinct subgroup that exhibits an abundance of potentially exploitable oncogenic drivers. These drivers include ERBB inhibitors such as afatinib and zenocutuzumab, TRK inhibitors like larotrectinib and entrectinib, the ALK/ROS inhibitor crizotinib, and the ongoing development of BRAF/MEK inhibitors. Within a limited population of individuals harboring the KRASG12C mutation, there is ongoing exploration into the therapeutic potential of the KRASG12C inhibitor known as AMG510, alongside other pharmacological compounds [12]. It is advisable to perform regular examinations for microsatellite instability (MSI) or deficient mismatch repair (dMMR) in patients with PDAC who are eligible for checkpoint inhibitor therapy. While infrequent in PDAC, individuals with tumors exhibiting microsatellite instability high (MSI-H) may experience significant advantages from the administration of programmed cell death protein (PD-1) inhibitor pembrolizumab. Consequently, it is advised to consider pembrolizumab as a potential treatment option for patients in the second-line or subsequent stages of therapy [253,267]. Pembrolizumab is a pharmacological intervention employed in the treatment of medical diseases defined by mismatch repair deficiency and microsatellite instability [268,269]. As previously elucidated, Table 4 provides a comprehensive summary of the existing pharmacological agents and their corresponding targets employed in the treatment of PDAC. The data presented in Table 4 elucidate the prevailing patterns in the utilization of chemotherapy.

## 2. Conclusions

As previously elucidated, the timely identification of PDAC holds paramount importance in enhancing the prognosis and overall well-being of afflicted individuals. The intriguing observation that a considerable proportion of patients diagnosed with PDAC present with the onset of diabetes mellitus offers a valuable avenue to identify a subgroup of individuals at heightened risk for developing PDAC [114]. The burgeoning significance of T2D in the pathogenesis of PDAC and the limited yet intriguing evidence regarding their prospective utility as diagnostic indicators and innovative therapeutic agents are indeed captivating. In light of the growing acknowledgement of the stroma’s significance in the advancement of cancer, it would be imprudent to disregard the potential impact of stromal factors in the context of PCRD [270]. Such studies are essential for unraveling the underlying mechanisms of PCRD and establishing a comprehensive understanding of the intricate relationship between PCRD and the advancement of cancer [271]. Future research in the field of PDAC treatment should prioritize investigations into the potential synergistic effects of established anti-cancer drugs, as well as studies into the efficacy of natural products in mitigating the adverse effects associated with anti-cancer drug therapies [221]. Thus, naringenin, aRVS, mistletoe, XLP, QYHJ, KLTi, and Korean red ginseng are currently being investigated for their potential therapeutic applications. It is anticipated that through demanding research and experimentation, these treatments may be further developed and contribute to the therapeutic options for cancer patients in the future. Notably, preliminary report presents findings from a randomized clinical trial investigating the effects of red ginseng on immunological control. The results of this study have garnered significant interest, highlighting the need for more research in this area. Such scientific pursuits possess the inherent capacity to unveil hitherto unexplored biomarkers and therapeutic targets, thereby facilitating the prompt detection and improved handling of this profoundly incapacitating ailment. The overarching objective is to effectively identify and encompass a substantial proportion of individuals afflicted with respectable, early stage PDAC who are presently being diagnosed belatedly.

## Figures and Tables

**Figure 1 ijms-24-15906-f001:**
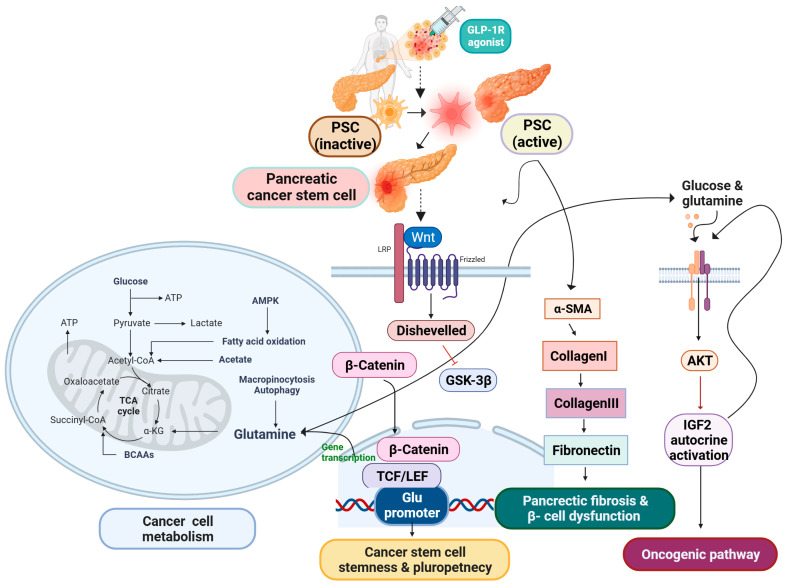
PSCs have emerged as a promising candidate for stem cells in PDAC. GLP-1R agonists possess the capacity to trigger the activation of PSCs situated within individuals diagnosed with diabetes [65]. The Wnt signaling pathway is known to be activated by PSCs. Wnt can directly bind to the GS promoter by connecting with the β-catenin/TCF7 complex. The inactive GSK3β can activate the signaling cascade, leading to increased GS expression within the promoter region. Glutamine metabolism has been observed to exert inhibitory effects on the processes of self-renewal and the expression of stemness/pluripotent genes [37]. The promotion of oncogenic Akt signaling is facilitated by the release of IGF2 through the action of glutamine. The production of IGF2 by beta cells is regulated by nutrients, which in turn activates the autocrine loop involving IGF2 and IGF1R. This loop plays a crucial role in controlling both the bulk and function of the β cells [43]. The activation of PSCs induces a cellular transformation that resembles the characteristics of myofibroblasts. This transformation is characterized by the production of α-smooth muscle actin (α-SMA), which serves as a protein marker for cellular activation. The process of activation serves to decrease the presence of fat droplets that contain retinoids. PSCs are of utmost importance in the intricate process of islet fibrosis formation, ultimately resulting in the impairment of β-cell functionality [51].

**Table 1 ijms-24-15906-t001:** The investigation of the potential anti-type 2 diabetic mellitus (T2DM) properties exhibited by natural compounds.

Compound	Mechanism	Experimental Model	Reference
Epicatechin and quercetin (1:1 combination)	reductions in total cholesterol, LDL-cholesterol, total triglycerides, and fasting plasma glucose	randomized placebo-controlled trial	[127]
Curcumin	nicotinamide adenine dinucleotide phosphate (NADPH) oxidases, which leads to an increase in the activity of antioxidant enzymes	healthy horses (mixed breeds, mean age 6.2 ± 2.3)	[128,129]
Combination of swertiamarin and quercetin (CSQ)	activation of viable β cells, reducing levels of total cholesterol, triglycerides, and LDL, while simultaneously increasing HDL	diabetic rats	[140]
Metformin	insulin or sulfonylureas	patients	[131,132]
Naringenin	upregulation of adenosine-monophosphate-activated protein kinase (AMPK), a key enzyme involved in cellular energy regulation.		[141]

**Table 2 ijms-24-15906-t002:** The in vitro investigation of the anti-cancer effect of natural products.

Compound	Mechanism	Experimental Model	Reference
*Abies spectabilis*	100% cell death at 50 μg/mL	MIA PaCa-2 cells	[150]
Apigenin	Increased the expression of the cytokine genes *IL17F*, *LTA*, *IL17C*, *IL17A*, and *IFNB1*	BxPC-3, PANC-1, BxPC-3 cells	[152]
Berberine	Release of cytochrome c and caspase 7	BxPC-3, HPDE-E6E7c7 cells	[154]
Coix seed extract	Inhibion of ABCB1 and ABCG2, combined with gemcitabine (combination index = 0.54) IC_50_ and tumor growth inhibition rate (TGI), bioluminescent pharmacokinetics	PANC-1, BxPC3^luc^ cells, BxPC3^luc^ xenografts nude mice	[156]
Combination of docetaxel or gemcitabine with curcumin	Metastasis suppression using a matrigel-transwell test and wound healing Downregulation of MMP2/MMP9 and N-cadherin proteins and the overexpression of TIMP1/TIMP2	PANC-1, HPAF-II, MIAPaCa-2 cells	[157]
Combination of gemcitabine with ethanolic extract of *Orthosiphon stamineus*	Inhibiting migration triggered caspase-3, causing early apoptosis by cleaved PARP	PANC-1 cells	[182]
*Eucalyptus globulus* Labill	Reduced cell viability	PANC-1 cells	[165]
*Eryngium Billardieri*	Reduction of cyclin D1 expression but increase in Bax leads to apoptosis no effect on normal cells	PANC-1, KDR/293 cells	[159]
Hernandezine	Increasing AMPK phosphorylation or ROS and lowering mTOR/p70S6K phosphorylation	Capan-1, SW1990 cells	[160]
*Kaempferia parviflora*	Anti-austerity efficacy Inhibiting colony formation	PANC-1 cells	[168]
Lycopene	Inhibiting NF-kB activity and reducing reactive oxygen species The ratio of Bax to Bcl-2 is increased, but active caspase-3 and cIAP1 and cIAP2 are downregulated	PANC-1 cells	[172]
Naringenin	Increased activity of ASK1, P38, P53, JNK, and reactive oxygen species Gemcitabine’s ability to prevent cancer drug resistance also limits tumor cell invasion	SNU-213, AsPC-1, PANC-1 cells	[211]
Combination with naringenin	Effects that suppress cell proliferation, invasion, and p38 signaling	MIA PaCa-2, PANC-1, SNU-213, BALB/c nude mice	[213]
*Obtriangulata* methanol extract	Accumulation of apoptotic effects in the G2/M phase due to ERK, Src, and STAT3 downregulation	BxPC3 cells	[179]
Panax notoginseng Saponins	Apoptosis by flow cytometry, colony formation, and EdU for cell proliferation; wound healing and transwell for cell migration and invasiveness; CCK-8 for cell viability	MIA PaCa-2, PANC-1 cells	[183]
*Portulaca oleracea*	Induction of apoptosis as decreasing CDK1 and increasing P53 gene expression	PANC-1, HUVEC cells	[187]
Pomegranate extract	Using a chick chorioallantoic membrane (CAM) model to examine the antiangiogenic effect	Suit-2 cells	[188]
QYHJ	Keap1/Nrf2/HO-1/NQO1 axis, p-PI3K/p-AKT/p-mTOR, and p-AKT/mTOR inhibition	PANC-1, MIA PaCa-2 cells	[176]
Quercetin	Reductions in EMT, invasion, and metastasis with downregulation of N-cadherin, MMP-9, and STAT-3	PANC-1 cells	[191]
*Rhus verniciflua* Stokes	Receptor DR3 mediates NF-kappaB pIkBalpha/beta kinases (pIKKs), MMP9, and XIAP downregulation	PaC cells	[194]
Shikonin	PAK1 inhibitors	BxPC−3, PANC−1 cells	[200]
Total flavonoid aglycone extracted	Inhibition of the Akt/mTOR/PI3K signaling cascade Cleavage of caspase proteins as part of the caspase signaling cascade in annexin V/PI-positive cells but showed low cytotoxicity toward normal cells	BxPC3, HPDE6-C7 cells	[195]
Triptolide	Inducing caspase-independent apoptosis Induces autophagic death	MiaPaCa-2, Capan-1, BxPC-3, S2-013, S2-VP10, Hs766T cells	[202]
Thymoquinone	Increased p21 and p53; downregulation of Bcl-xL, Bcl-2, XIAP, Notch1, NICD, PTEN, Akt/mTOR/S6 signaling Upregulation of caspases-3, -9, Bax, and cytochrome c; G2 cycle arrest and Sub G0/G1 arrest	MiaPaCa-2, AsPC-1 cells	[206,207,208,209]
Wogonin	p53 expression was stimulated, and the Beclin-1/PI3K, Akt/ULK1/4E-BP1/CYLD, and mTOR pathways were all inhibited	PANC-1, Colo-357, HPCCs4, Capan-1, Colo-357 cells	[144]
Wogonin	Suppressing Mcl-1, CDK-9, c-FLIP, and MDM2	Capan-1, Colo-357 cells	[215,216]
XLP	A reduction in the activity of MMP2, PTGS2, CASP9, IL4, and CTSD	Mouse PC Panc-02 cell transfection pCDNA3.1(+)-PTGS2 and pCDNA3.1(+)-NC, male C57BL/6 mice, 4–6 weeks old	[177,222]
*Ziziphus nummularia* ethanolic extract	Suppresses of MMP-9, proliferation, migration, invasive potential, ERK1/2(MAPK), NFκB pathways, collagen adhesion, integrin α2 expression, and VEGF production Induction of caspase-3-dependent	Capan-2 cells	[219]

**Table 3 ijms-24-15906-t003:** Investigating the effectiveness of natural products against PDAC in the clinic.

Compound	Efficacy	Reference
Curcuminoids	Health benefits, including fewer unwanted consequences, are increasing	[238,239,240]
Mistletoe (*Viscum album*)	Improve survival rates from the current 2.7 months to 4.8	[233,235]
QYHJ	Improved 1-, 3-, and 5-year survival with no obvious side effects	[177]
aRVS	Improve survival rates with single or combined treatment	[228]
Combination of curcuminoids and gemcitabine	Curcumin increases chemotherapy’s efficacy	[237]
Combination of kanglaite injection (KLTi) and gemcitabine	Improve survival rates with quality of life Or no obvious side effects	[229]
Korean red ginseng	Elevated CD4+ lymphocytes and a higher CD4+/CD8+ T lymphocyte counts	[249,250]

**Table 4 ijms-24-15906-t004:** An Investigation on the potential of a drug as a therapeutic intervention for PDAC.

Drug	Target	Characterization	Reference
Olaparib	PDAC drivers with KRAS mutations, NTRK1–3 fusions, and BRCA1/2 mutations	BRCA1/2 mutations	[266]
Afatinib and zenocutuzumab	PDAC drivers, which are characterized by the presence of wild-type KRAS, include ERBB inhibitors	ERBB mutations	[12]
Larotrectinib and entrectinib	PDAC drivers include TRK inhibitors, which are activated by the presence of wild-type KRAS	TRK mutations	[12]
Crizotinib	PDAC drivers with wild-type KRAS include sensitivity to ALK/ROS inhibitors	ALK/ROS mutations	[12]
Pembrolizumab	Programmed cell death protein (PD-1) inhibitor	Microsatellite instability high (MSI-H)	[253,267]
FOLFIRINOX; leucovorin (LV), 5-fluorouracil (5-FU), irinotecan, and oxaliplatin	C-X-C motif chemokine 5 (CXCL5)	Influence on tumor immunity regulation	[259]

## Data Availability

Not applicable.

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
