# Peer review of "Therapeutic Strategies for Pancreatic-Cancer-Related Type 2 Diabetes Centered around Natural Products"

_ijms, 2023, doi:10.3390/ijms242115906_

Round 1

Reviewer 1 Report

Comments and Suggestions for Authors

Dear Author,

your manuscript "Therapeutic strategies for Pancreatic ductal adenocarcinoma are centered on natural product"  is well structured and very interesting.

For these reasons, I think that it can be accept in this form.

Best regards

Author Response

Dear Author,

your manuscript "Therapeutic strategies for Pancreatic ductal adenocarcinoma are centered on natural product"  is well structured and very interesting.

For these reasons, I think that it can be accept in this form.

Best regards

>> (Response) I am grateful for your valuable time and kind words of support.

I hope your research and everything else goes well.

Warm regards.

Moon Nyeo Park

Reviewer 2 Report

Comments and Suggestions for Authors

1.     The review focused on natural product-based therapeutic strategies for PDAC. However, I found another interesting perspective in this article, which is a correlation between type-II diabetes and PDAC development. The audience couldn’t get any hints on this from the title of the review. 

2.     I recommend modifying the introduction part like a constructive story connecting all the main objectives of this review. At the end of the introduction part, the author should clearly mention all the aims of the review. The author should include the relevant and recent references in this section.

3.     I found section numbering only for the introduction and conclusion. The other topic sentences in the body should be numbered as well.

4.     It would be great if the author can provide a pictorial representation connecting Type-II diabetes and PDAC

5.     Under the section, ‘Diabetes management strategies may reduce the risk of pancreatic cancer’, the author should include other plant-based compounds such as Triterpenoids, saponin that play important roles in Type-II diabetes and PDAC management.

6.     The sections named ‘Herbal medicine in vitro assay’ and ‘Natural compound in clinical trial’ focus on PDAC. So, it is better to mention that clearly in the headings. The author can mention the role of another flavonoid, apigenin in PDAC management under the section ‘Herbal medicine in vitro assay’. This section also describes the effects of herbal medicine in vivo systems as well. So, I suggest to change this heading.

7.     The sections named ‘Tendency in FDA-Approved Therapies’ focused on nanotherapy, immunotherapy, and targeted therapy. At the end of this section, I can see Table 4 on different chemo drug options. In my opinion, the author can start with chemo drugs first. However, the author should write this section with proper references.

8.     I found some typos such as in line no. 181, 330. The author should be careful with sentence construction such as line no. 53-55. I recommend authors should carefully read the whole manuscript and also modify the overall language quality.

Comments on the Quality of English Language

I found some typos such as in line no. 181, 330. Author should be careful on sentence construction such as line no. 53-55. I recommend authors should carefully read the whole manuscript and also modify the overall language quality.

Author Response

Comments and Suggestions for Authors

  1. The review focused on natural product-based therapeutic strategies for PDAC. However, I found another interesting perspective in this article, which is a correlation between type-II diabetes and PDAC development. The audience couldn’t get any hints on this from the title of the review. 

I sincerely appreciate the Reviewer’s advice I missed to consider from every part of my heart. As Reviewer pointed out, I put in every effort to revise it. Once more, I greatly appreciate your time and the advice you provided regarding my thesis.

>> (Response) I modified title such as Therapeutic strategies for the Pancreatic cancer related type II diabetes are centered around natural products.

  1. I recommend modifying the introduction part like a constructive story connecting all the main objectives of this review. At the end of the introduction part, the author should clearly mention all the aims of the review. The author should include the relevant and recent references in this section.

>> (Response) I modified page 2 line 72-85

  1. I found section numbering only for the introduction and conclusion. The other topic sentences in the body should be numbered as well.

>> (Response) I will plan to address the additional specifics of this subject in more detail in the following paper, therefore I altered the subtitle.

  1. It would be great if the author can provide a pictorial representation connecting Type-II diabetes and PDAC

>> (Response) I sincerely appreciate it. I was able to modify it from a more objective standpoint because of the advice. Thank you one more. I modified page 5, Figure 1

  1. Under the section, ‘Diabetes management strategies may reduce the risk of pancreatic cancer’, the author should include other plant-based compounds such as Triterpenoids, saponin that play important roles in Type-II diabetes and PDAC management.

>> (Response) I've simply highlighted representative items thus far, but I’ll be going to a lot of worthy candidates under other topics.

  1. The sections named ‘Herbal medicine in vitro assay’ and ‘Natural compound in clinical trial’ focus on PDAC. So, it is better to mention that clearly in the headings. The author can mention the role of another flavonoid, apigenin in PDAC management under the section ‘Herbal medicine in vitro assay’. This section also describes the effects of herbal medicine in vivo systems as well. So, I suggest to change this heading.

>> (Response) I modified, page 9 line 354

>> (Response) I modified, page 11 line 438

  1. The sections named ‘Tendency in FDA-Approved Therapies’ focused on nanotherapy, immunotherapy, and targeted therapy. At the end of this section, I can see Table 4 on different chemo drug options. In my opinion, the author can start with chemo drugs first. However, the author should write this section with proper references.

 >> (Response) I modified, page 14 Table 4

  1. I found some typos such as in line no. 181, 330. The author should be careful with sentence construction such as line no. 53-55. I recommend authors should carefully read the whole manuscript and also modify the overall language quality.

>> (Response) I modified page 2 line 54, page 5 line 190, page 8 line 348.

Reviewer 3 Report

Comments and Suggestions for Authors

The manuscript has many great points that are very useful and have the potential to become better. Throughout the manuscript, there is, however, a recurrent problem of not giving the citations where necessary and random abbreviations as highlighted below:

Major comments:

-       On page 3, line 126, please provide a reference.

-       What are SP cells in line 142, on the same page?

-       There is only a single figure with a very small font size. The author must create a figure to depict various natural compounds discussed in the section(s) and how they target various pathways to curtail or inhibit pancreatic tumor growth (or PCRD) or metastasis.

-       On page 5, line 227, the authors state that – “Obesity and T2D are linked by reduced β cell ability to overcome insulin resistance.” please provide reference for the obesity angle.

-       Correct grammar and make the sentence more meaningful on page 5, line 224 – “Research findings have shown that CD36 has a role in that infiltrate tumors, leading to the occurrence of lipid peroxidation.”

-       On line 228, the author states that – “It also increases systemic inflammation..” What is “it” here?

-       Provide a reference for the statement on lines 239-240 - “Following the occurrence of an injury, immune cells are recruited to the affected site and subsequently secrete several cytokines, such as IL-1, TNF alpha, and IL-6.”

-       On line 241, the authors state that – “Additionally, growth factors including TGF β 1, FGF2, and PDGF angiotensin II are released[91].” Released by what and why is it relevant? Also, there is a major problem in the cited reference as there is no mention of “angiotensin II” in the cited reference. This is misleading. Please correct!

-       G

-        On line 256, the authors state that - “Therefore, the prevention of obesity-induced activation of the insulin-IGF-1 axis represents a promising therapeutic target for cancer treatment [97].” This cited study is not for pancreatic cancer. In the literature, there are plenty of papers relating obesity, IGF-1, and pancreatic cancer.

-       Provide a reference for the statement on lines 308-309.

-       What is OOE on line 393?

-       On line 399, provide a sentence or two as a background on these natural compounds before discussing them such as – thymoquinone, berberine, etc.

-       Metformin is NOT a Natural compound. The source of metformin is galegine, a natural product produced by the plant Galega officinalis. Please correct in table 2 and line 383 ( and other places).

-       In the section, “Natural compound in clinical trial”, the authors must include triptolide/minnelide (https://www.sciencedirect.com/science/article/abs/pii/S0304383522000660). This is currently in Phase II clinical trial for advanced pancreatic cancer.

-       On line 442, correct “standalone” to stand-alone.

-       Provide background for “mistletoe” in line 443.

-       On line 498, correct “D+/CD8+).

-       Provide a reference for the first sentence in line 508.

Minor comments:

-       On page 2, line 54, correct grammar and complete the sentence (“Whilst” should be in small letters).

Comments on the Quality of English Language

Heavy edits are required as mentioned above.

Author Response

Comments and Suggestions for Authors

The manuscript has many great points that are very useful and have the potential to become better. Throughout the manuscript, there is, however, a recurrent problem of not giving the citations where necessary and random abbreviations as highlighted below:

 I sincerely appreciate the Reviewer’s advice I missed to consider from every part of my heart. As Reviewer pointed out, I put in every effort to revise it. Once more, I greatly appreciate your time and the advice you provided regarding my thesis.

Major comments:

-       On page 3, line 126, please provide a reference.

 >> (Response) I modifed page 3 line 128.

-       What are SP cells in line 142, on the same page?

>> (Response) I modifed page 3 line 144.

-       There is only a single figure with a very small font size. The author must create a figure to depict various natural compounds discussed in the section(s) and how they target various pathways to curtail or inhibit pancreatic tumor growth (or PCRD) or metastasis.

>> (Response) I will plan to address the additional specifics of this subject in more detail in the following paper.

-       On page 5, line 227, the authors state that – “Obesity and T2D are linked by reduced β cell ability to overcome insulin resistance.” please provide reference for the obesity angle.

>> (Response) I modifed page 6 lin2 242-243.

-       Correct grammar and make the sentence more meaningful on page 5, line 224 – “Research findings have shown that CD36 has a role in that infiltrate tumors, leading to the occurrence of lipid peroxidation.”

 >> (Response) I modifed page 6 lin2 234-235.

-       On line 228, the author states that – “It also increases systemic inflammation.” What is “it” here?

 >> (Response) I modifed additional information , page 6 lin2 237-243.

-       Provide a reference for the statement on lines 239-240 - “Following the occurrence of an injury, immune cells are recruited to the affected site and subsequently secrete several cytokines, such as IL-1, TNF alpha, and IL-6.” 

>> (Response) I modifed, page 6 line 254.

-       On line 241, the authors state that – “Additionally, growth factors including TGF β 1, FGF2, and PDGF angiotensin II are released[91].” Released by what and why is it relevant? Also, there is a major problem in the cited reference as there is no mention of “angiotensin II” in the cited reference. This is misleading. Please correct!   ” 

>> (Response) I modifed, page 6 line 254-256.

-        On line 256, the authors state that - “Therefore, the prevention of obesity-induced activation of the insulin-IGF-1 axis represents a promising therapeutic target for cancer treatment [97].” This cited study is not for pancreatic cancer. In the literature, there are plenty of papers relating obesity, IGF-1, and pancreatic cancer.

>> (Response) I modifed page 7 line 271-275.

-       Provide a reference for the statement on lines 308-309.

>> (Response) I modifed page 8 line 328.

-       What is OOE on line 393?

>> (Response) I modifed page 9 line 400.

-       On line 399, provide a sentence or two as a background on these natural compounds before discussing them such as – thymoquinone, berberine, etc. –

>> (Response) I modifed page 9 line 406-408.

-       Metformin is NOT a Natural compound. The source of metformin is galegine, a natural product produced by the plant Galega officinalis. Please correct in table 2 and line 383 ( and other places).

>> (Response) I modified, page 10 line 483 table 2.

-       In the section, “Natural compound in clinical trial”, the authors must include triptolide/minnelide (https://www.sciencedirect.com/science/article/abs/pii/S0304383522000660). This is currently in Phase II clinical trial for advanced pancreatic cancer.

>> (Response) I modified additional information according to reviewer’s recommendations, page 11 line 483-487.

-       On line 442, correct “standalone” to stand-alone. 

>> (Response) I modified page 11 line 448.

 -       Provide background for “mistletoe” in line 443.

>> (Response) I modified page 11 line 448-450.

-       On line 498, correct “D+/CD8+).

>> (Response) I modified page 12 line 519.

-       Provide a reference for the first sentence in line 508.

 >> (Response) I modified page 13 line 529-532.

Minor comments:

-       On page 2, line 54, correct grammar and complete the sentence (“Whilst” should be in small letters).

>> (Response) I modified page 2 line 56.

Round 2

Reviewer 2 Report

Comments and Suggestions for Authors

Author has improved the manuscript as suggested.

Author Response

I am truly grateful for your expert guidance that helped my manuscript to be improved in quality.

I hope all of your future endeavors and research are filled with success.

Warm regards

Moon Nyeo Park

Reviewer 3 Report

Comments and Suggestions for Authors

The manuscript has many great points that are very useful and have the potential to become better. Throughout the manuscript, there is, however, a recurrent problem of not giving the citations where necessary and random abbreviations as highlighted below:

Response: I sincerely appreciate the Reviewer’s advice I missed to consider from every part of my heart. As Reviewer pointed out, I put in every effort to revise it. Once more, I greatly appreciate your time and the advice you provided regarding my thesis.

Response: The authors must comply and put in more effort. IJMS is a reputed journal and frankly, many of the queries posted here will be raised by any journal!

Major comments:

-       On page 3, line 126, please provide a reference.

 >> (Response) I modifed page 3 line 128.

Query: The authors were asked about the reference on line 126 because there were two scientific claims made with no references – “PSCs have the ability to undergo activation, resulting in the acquisition of a myofibroblast-like phenotype characterized by the expression of α-smooth muscle actin (α-SMA) as an activation marker protein (ref. 48,49 – authors provided). This activation process is accompanied by a decrease in the quantity of retinoid-containing fat droplets (ref??). The authors did NOT provide a reference for the second sentence on the same line in the original document (line 126). Please provide references. Also, reference 48 provided by the author is wrong and has no association with the activation of pancreatic stellate cells and the acquisition of myofibroblast phenotype. Please comply.

-       What are SP cells in line 142, on the same page?

>> (Response) I modifed page 3 line 144.

Query: When the authors were asked about the SP cells, we were not only looking for a mere abbreviation. Some level of background must be provided before mentioning a new cell type. This goes even for the nestin-positive cells. This is okay for scientists in the field but for a larger audience, this is very important.

-       There is only a single figure with a very small font size. The author must create a figure to depict various natural compounds discussed in the section(s) and how they target various pathways to curtail or inhibit pancreatic tumor growth (or PCRD) or metastasis.

>> (Response) I will plan to address the additional specifics of this subject in more detail in the following paper.

Query: The font size of the figure components for example axin, CK1alpha, GSK-3beta, etc. are very small. What is more, is that they have not been discussed anywhere in the manuscript. In addition, the authors must realize that the paper is titled and centered around the core that “Therapeutic strategies for Pancreatic ductal adenocarcinoma are centered on natural product”. It is therefore essential for a figure to be there to depict various natural compounds discussed in the section(s) and how they target various pathways to curtail or inhibit pancreatic tumor growth (or PCRD) or metastasis.

-       On page 5, line 227, the authors state that – “Obesity and T2D are linked by reduced β cell ability to overcome insulin resistance.” please provide a reference for the obesity angle.

>> (Response) I modifed page 6 lin2 242-243.

Query: Authors state that – “Diabetes type 2 is associated with obesity because cells are unable to compensate for insulin resistance [97].” This must be explained better. Which cell type are the authors talking about beta? Mention. Provide the actual reference that demonstrates that Type 2 diabetes is linked to obesity because cells are unable to compensate for insulin resistance. This is not provided in the reference cited.

-       What is OOE on line 393?

>> (Response) I modifed page 9 line 400.

Query: As the authors have explained Thymoquinone (Tq) or Qingyihuaji (QYHJ), they must briefly explain OOE as well. A mere abbreviation will not suffice.

-       Metformin is NOT a Natural compound. The source of metformin is galegine, a natural product produced by the plant Galega officinalis. Please correct in table 2 and line 383 ( and other places).

>> (Response) I modified, page 10 line 483 table 2.

Query: On lines 486-487, the author states that – “Metformin, classified as a biguanide, is a commonly prescribed pharmaceutical agent utilized for the management of T2DM [190].” Metformin IS NOT A NATURAL COMPOUND. The heading for the section and table 3, indicates natural products or compounds. Why can the authors not change it to something like galegine-derived Metformin?

Comments on the Quality of English Language

Fine

Author Response

Comments and Suggestions for Authors

The manuscript has many great points that are very useful and have the potential to become better. Throughout the manuscript, there is, however, a recurrent problem of not giving the citations where necessary and random abbreviations as highlighted below:

Response: I sincerely appreciate the Reviewer’s advice I missed to consider from every part of my heart. As Reviewer pointed out, I put in every effort to revise it. Once more, I greatly appreciate your time and the advice you provided regarding my thesis.

Response: The authors must comply and put in more effort. IJMS is a reputed journal and frankly, many of the queries posted here will be raised by any journal!

Major comments:

-       On page 3, line 126, please provide a reference.

 >> (Response) I modified page 3 line 128.

QueryThe authors were asked about the reference on line 126 because there were two scientific claims made with no references – “PSCs have the ability to undergo activation, resulting in the acquisition of a myofibroblast-like phenotype characterized by the expression of α-smooth muscle actin (α-SMA) as an activation marker protein (ref. 48,49 – authors provided). This activation process is accompanied by a decrease in the quantity of retinoid-containing fat droplets (ref??). The authors did NOT provide a reference for the second sentence on the same line in the original document (line 126). Please provide references. Also, reference 48 provided by the author is wrong and has no association with the activation of pancreatic stellate cells and the acquisition of myofibroblast phenotype. Please comply.

>> (Response) I modified page 3 line 128, and 130.

-       What are SP cells in line 142, on the same page?

>> (Response) I modified page 3 line 144.

QueryWhen the authors were asked about the SP cells, we were not only looking for a mere abbreviation. Some level of background must be provided before mentioning a new cell type. This goes even for the nestin-positive cells. This is okay for scientists in the field but for a larger audience, this is very important.

>> (Response) I modified page 3-4, line 144, and 148.

-   There is only a single figure with a very small font size. The author must create a figure to depict various natural compounds discussed in the section(s) and how they target various pathways to curtail or inhibit pancreatic tumor growth (or PCRD) or metastasis.

>> (Response) I will plan to address the additional specifics of this subject in more detail in the following paper.

Query: The font size of the figure components for example axin, CK1alpha, GSK-3beta, etc. are very small. What is more, is that they have not been discussed anywhere in the manuscript.

>> (Response) I modified page 3 line 106-108 and page 5 line 192-194 as well as the font size.

 In addition, the authors must realize that the paper is titled and centered around the core that “Therapeutic strategies for Pancreatic ductal adenocarcinoma are centered on natural product”. It is therefore essential for a figure to be there to depict various natural compounds discussed in the section(s) and how they target various pathways to curtail or inhibit pancreatic tumor growth (or PCRD) or metastasis.

>> (Response) I extensively modified page 9 line 395-405, page 9-10 line 410-449, page 11-12 line 475-530, page 12 line 557-565, page 15 line 585-590, page 16 line 632-636, table 2 page 13, page 14, page 15 and table  3 page 17.

-       On page 5, line 227, the authors state that – “Obesity and T2D are linked by reduced β cell ability to overcome insulin resistance.” please provide a reference for the obesity angle.

>> (Response) I modifed page 6 lin2 242-243.

Query: Authors state that – “Diabetes type 2 is associated with obesity because cells are unable to compensate for insulin resistance [97].” This must be explained better. Which cell type are the authors talking about beta? Mention. Provide the actual reference that demonstrates that Type 2 diabetes is linked to obesity because cells are unable to compensate for insulin resistance. This is not provided in the reference cited.

>> (Response) I modified page 6 lin2 250-265.

-       What is OOE on line 393?

>> (Response) I modifed page 9 line 400.

Query: As the authors have explained Thymoquinone (Tq) or Qingyihuaji (QYHJ), they must briefly explain OOE as well. A mere abbreviation will not suffice.

>> (Response) I modified page 10-11 line 467-469. 

-       Metformin is NOT a Natural compound. The source of metformin is galegine, a natural product produced by the plant Galega officinalis. Please correct in table 2 and line 383 ( and other places).

>> (Response) I modified, page 10 line 483 table 2.

Query: On lines 486-487, the author states that – “Metformin, classified as a biguanide, is a commonly prescribed pharmaceutical agent utilized for the management of T2DM [190].” Metformin IS NOT A NATURAL COMPOUND. The heading for the section and table 3, indicates natural products or compounds. Why can the authors not change it to something like galegine-derived Metformin?

>> (Response) Thank you for kind consideration. According to your suggestion, Metform has been removed from sections and Table 3.
